# Complexity Analysis of EEG, MEG, and fMRI in Mild Cognitive Impairment and Alzheimer’s Disease: A Review

**DOI:** 10.3390/e22020239

**Published:** 2020-02-20

**Authors:** Jie Sun, Bin Wang, Yan Niu, Yuan Tan, Chanjuan Fan, Nan Zhang, Jiayue Xue, Jing Wei, Jie Xiang

**Affiliations:** College of Information and Computer, Taiyuan University of Technology, Taiyuan 030024, China; sj13834650566@163.com (J.S.); wangbin01@tyut.edu.cn (B.W.); niuyan0049@link.tyut.edu.cn (Y.N.); tanyuan0339@link.tyut.edu.cn (Y.T.); fanchanjuan0303@link.tyut.edu.cn (C.F.); zhangnan0326@link.tyut.edu.cn (N.Z.); xuejiayue0062@link.tyut.edu.cn (J.X.); 20141032@sxufe.edu.cn (J.W.)

**Keywords:** Alzheimer’s disease, complexity, brain signals, single-channel analysis, biomarker

## Abstract

Alzheimer’s disease (AD) is a degenerative brain disease with a high and irreversible incidence. In recent years, because brain signals have complex nonlinear dynamics, there has been growing interest in studying complex changes in the time series of brain signals in patients with AD. We reviewed studies of complexity analyses of single-channel time series from electroencephalogram (EEG), magnetoencephalogram (MEG), and functional magnetic resonance imaging (fMRI) in AD and determined future research directions. A systematic literature search for 2000–2019 was performed in the Web of Science and PubMed databases, resulting in 126 identified studies. Compared to healthy individuals, the signals from AD patients have less complexity and more predictable oscillations, which are found mainly in the left parietal, occipital, right frontal, and temporal regions. This complexity is considered a potential biomarker for accurately responding to the functional lesion in AD. The current review helps to reveal the patterns of dysfunction in the brains of patients with AD and to investigate whether signal complexity can be used as a biomarker to accurately respond to the functional lesion in AD. We proposed further studies in the signal complexities of AD patients, including investigating the reliability of complexity algorithms and the spatial patterns of signal complexity. In conclusion, the current review helps to better understand the complexity of abnormalities in the AD brain and provide useful information for AD diagnosis.

## 1. Introduction

Alzheimer’s disease (AD) is the most prevalent form of neurodegenerative dementia and includes a set of symptoms, such as memory loss and cognitive decline, that affect the ability to engage in daily activities and processes, including attention, thinking, orientation, or language [1,2]. In AD patients, proteins accumulate in the brain, forming amyloid plaques and neurofibrillary tangles, which have been shown to be associated with local synaptic disruptions [3,4]. Eventually, AD leads to the loss of connections between nerve cells, suggesting that AD is a disconnectivity disease. There are currently two recognized predementia stages: subjective cognitive impairment (SCI) and mild cognitive impairment (MCI) [5,6]. SCI refers to an individual’s main complaint of cognitive impairment with a lack of objective evidence of cognitive impairment or pathology. In recent years, SCI has become a hot topic in the research field of AD [5,7]. MCI increases the risk of and is an important risk factor for AD dementia, thus becoming an important target for early diagnosis of and intervention for AD [6]. Both SCI and MCI patients are at great risk of developing AD. Therefore, an in-depth understanding of the mechanisms involved in the early diagnosis and effective treatment of AD is crucial.

Brain imaging analyses have been widely used to explore the mechanisms of AD [8,9,10] and improve the accuracy of AD diagnosis [11,12]. Because the brain is a highly complex system and brain signals have complex nonlinear dynamics, there has been increasing interest in complexity analyses by using brain imaging data such as electroencephalograms (EEG), magnetoencephalogram (MEG), and functional magnetic resonance imaging (fMRI) [13,14,15]. Most studies have analyzed brain signals from a single channel, such as the signals from an electrode in EEG, a channel in MEG, or a voxel in fMRI. Recently, the complexity of brain signals has been widely used to better understand the complexity of abnormalities in the AD brain. Adequate study of brain imaging modalities provides an opportunity to outline the mechanisms underlying AD and useful information for its diagnosis [16,17,18]. More recently, some studies have proposed that the levels of complexity are potential biomarkers for identification in the early diagnosis of AD [19,20]. To date, there is no comprehensive review that summarizes the different imaging modalities and explains the complexity of abnormalities in the AD brain.

In the present review, we systematically examined 126 identified studies on the complexity of AD from 2000 to 2019. We aim to review the complexity indexes that can accurately represent the functional lesion in AD and outline the better complexity indicators. In addition, by analyzing changes in patients through general trends and comparative studies of brain regions, we identified our knowledge gaps as well as new issues for future research that can serve as a starting point for future applications of complexity analysis for AD patients.

## 2. Methods

### 2.1. The Analysis of Complexity

Entropy (En) is one of the most commonly used nonlinear concepts in evaluating the dynamic characteristics of signals [21]. This concept is an index of complexity analysis reflecting the degree of system confusion in a time series. These methods combine the complexity of the signal with its unpredictability: irregular signals are more complex than regular ones because they are more unpredictable. Some researchers believe that these techniques can be used to analyze time series in the time domain or frequency domain. In the time domain, entropy mainly reflects the changes in time, and these analyses are constantly improving. Approximate entropy (ApEn) is an indicator of the overall characteristics of the response signal from the point of view of the complexity of the signal. It is useful for small datasets and is effective for discriminating the signal from random signals [22,23]. Then, this index was replaced by sample entropy (SampEn), introduced by Richman and Moorman [24]. The sample entropy algorithm does not include a comparison to its own data; it is the exact value of the negative average natural logarithm of the conditional probability and has good consistency [25]. Fuzzy entropy (FuzzyEn) uses the exponential fuzzy similarity measure formula, which is more stable than the sample entropy algorithm [26]. Permutation entropy (PeEn) is a method for measuring nonstationary time series irregularities. PeEn considers only the grades of the samples but not their metrics [27]. PeEn has certain advantages over the other commonly used entropy metrics, including its simplicity, low computational complexity without further model assumptions, and robustness in the presence of observations and dynamic noise [27,28]. It has been successfully applied to EEG analyses and has been reported to be a good biomarker for distinguishing normal elderly individuals from patients with MCI and AD [29,30]. However, these methods mostly consider features at a single scale and can reflect only one aspect of the brain signal. Researchers have argued that multiscale entropy-based approaches better reflect the gradual transition process from coarse-grained entropy to fine-grained entropy, which can well reflect the complexity of biological signals on different time scales.

Although they continue to be rigorous and widespread methods used in the analysis of the frequency domain, linear decomposition methods, such as spectral analysis, have recently been suggested to lead to a loss of unique information that is orthogonal to average activity [31,32]. Renyi entropy (ReEn) is a generalization of Shannon entropy (ShEn), collision entropy, and minimum entropy, and it quantifies the diversity, uncertainty, or randomness of the system. Renyi entropy forms the basis of the concept of generalized dimensionality [33,34]. Tsallis entropy (TsEn) is nonexpansive [35]. For a composite system composed of two independent subsystems, it is not a simple sum of the entropy of two systems [36,37]. Spectral entropy (SpecEn) was developed to quantify the flatness of a spectrum [36,38]. SpecEn characterizes the distribution of power spectral density (PSD) by assessing disorder in the spectrum.

In addition to the entropy method, there are many other methods for assessing complexity, such as the Hurst exponent (HE), the Lempel-Ziv complexity (LZC), the correlation dimension (D2), and the fractal dimension (FD). The HE is mainly used to measure the long-term memory and fractal dimension of a time series [39]. The LZC reconstructs the original time series into a binary sequence [40]. The D2 and the largest Lyapunov exponent (LLE) were the first nonlinear techniques applied to EEG and MEG signals [41,42]. However, the calculation of D2 and LLE requires the signals to be stationary and long enough [43,44], which cannot be achieved for physiological data [45,46]. The FD has proven to be a reliable indicator for identifying healthy and pathological brains, and it can track changes in the complexity of neuronal dynamics, which might be related to cognitive or perceptual impairments [47]. Higuchi’s fractal dimension (HFD) is a fast computational method for obtaining the FD of a time series signal [48], even when very few data points are available. In addition, HFD provides a more accurate way to measure signal complexity [49,50], and it has proven to be an effective way to distinguish between AD patients and normal subjects.

Table 1 briefly introduces some widely used complexity methods. Although there are a large number of methods to assess complexity, entropy is the most popular. There are some problems with these methods, such as missing information, sensitivity to noise, and inaccurate results. The entropy method is advantageous in that it requires only a small amount of analysis data, possesses a strong anti-interference ability, and involves a simple algorithm. Different complexity analysis methods have their own advantages and disadvantages, and in this paper summarize their use in the analysis of brain signals acquired by different modalities in AD.

### 2.2. Literature Search

We examined the use of complexity techniques in the brain imaging of AD patients by performing an overview of these studies. Preferred Reporting Items for Systematic Reviews and Meta-Analyses (PRISMA) [62] was used to identify studies and narrow the collection for this review. We performed a search on Web of Science and PubMed using the following group of keywords: (“Complexity analysis” OR “Nonlinear dynamical analysis” OR “Lempel-Ziv complexity” OR “fractal dimension” OR “Hurst exponent” OR “entropy” OR “correlation dimension”) AND (“Alzheimer’s disease” OR “Mild Cognitive Impairment” OR “Subjective Cognitive Impairment”). References from 2000 until 2019 were used for further analysis. As shown in Figure 1, after excluding unqualified studies, this review narrowed the original count of 382 studies to the final count of 126 studies. Studies were divided into three categories: EEG (64%), MEG (28%), and fMRI and functional near-infrared spectroscopy (fNIRS) (7%) (Figure 2A). Various methods have been developed to examine the different types of brain imaging modalities, so the current status of these studies will also be described in the corresponding sections below. Unsurprisingly, EEG data are widely used in nonlinear analyses, accounting for 64% of all identified studies (Figure 2A). The four most commonly used analysis methods in the reviewed articles were time-domain entropy (TD-En), frequency-domain entropy (FD-En), LZC, and FD. The trends in the number of different techniques used in these brain imaging studies are shown in Figure 2B.

## 3. Results

### 3.1. Complexity Analysis of EEG Signals in AD

A large number of nonlinear methods have been applied to analyze the characteristics of brain activity in patients with AD, and numerous interesting results have been found. Since resting-state data are not influenced by task-related activation or differences in motivation or performance, these recordings provide more reliable estimates of brain adaptability [2,63]. Recordings of resting brain activity and task-related recordings exhibit similar network dynamics [64,65], and resting states often reflect the contribution of networks with the most metabolic activity [66]. The EEG signal has the advantage of high time resolution [67], and we found that the signals have been mainly analyzed in different frequency bands and from electrodes to reflect the variation in different signal values [68].

#### 3.1.1. Complexity Analysis in Entropy

In this section, we review the signal complexity of the resting-state electroencephalogram (rsEEG) in SCI, MCI, and AD patients compared with normal controls (NCs). Several studies have shown that multiple complexity methods, such as LZC, entropy complexity, and other complexity features, differ among SCI, MCI, AD, and control subjects when applied to EEG signals. Hogan et al. [69] found that the entropy in MCI subjects was low. A recent study reported that in all channels, the complexity values of the EEG signals from AD patients were shown to be below those from SCI patients. It has been demonstrated that ApEn [70,71] and SampEn [72,73] in EEG signals are significantly reduced in MCI and AD patients compared to healthy individuals [74,75]; Garn et al. used different methods [76] to explore the complexity of EEG signals from AD patients and age-matched control subjects. In recent years, studies have included LZC, distance-based LZC [77], ApEn, SampEn, multiscale sample entropy (MSE), and FuzzyEn analyses [78]. Consistent results were found in the EEGs of patients with AD, including a significant reduction in complexity at electrodes P3, P4, O1, and O2 placed over the parietal, occipital, and temporal regions compared to healthy individuals. We found that at the MCI stage, the medial temporal lobe, associated with short-term memory, is affected, and the lateral temporal lobe and parietal lobe [79] are also affected. In the moderate stage of AD, the frontal lobe is affected. During the severe stage of AD, the occipital lobe is affected [18]. Multiple entropy methods have been used to study the brain states that develop in the transition from healthy conditions to AD. Most of the studies have focused on particular areas in the brain. Figure 3 presents comparative values of entropy shown over five regions in AD, MCI, and NC subjects. AD and MCI patients had lower En values in the five regions ($En_{AD}$ < $En_{MCI}$< $En_{Control}$), and significant differences were observed among the frontal, temporal, and central regions. These results suggest that the EEG signals in the brains of AD and MCI patients had significantly less complexity in the frontal, temporal, and central regions than those in the NC subjects. Furthermore, AD patients exhibit the lowest complexity and the greatest regularity. As expected, the complexity of the EEG signals gradually decreases with disease development, especially when comparing NC subjects with patients with AD.

We think that the reduction in the irregularity or complexity of brain signals can be described by a decrease in the dynamic complexity of the brain [80]. Our review demonstrated that aging and age-dependent diseases are frequently accompanied by losses in a broad range of physiological complexity or irregularity. A theory of discontinuous syndrome might explain the changes in AD: plaques and cell death can lead to the loss of connectivity between cortical neurons, which may lead to more regular brain signals (as recorded by cortical brain activity), thus destroying effective communication throughout the brain and producing the range of commonly seen AD symptoms.

#### 3.1.2. Complexity Analysis in Multiscale Entropy

Entropy-based MSE analyses can measure the probability of sequences generating new information at different scales and have been applied to cognitive neuroscience. Deng et al. [82] studied changes on a 1–8 scale using multiscale weighted permutation entropy and found that the entropy in AD patients was decreased in the temporal, top, and right frontal occipital to the top and left occipital regions. Mizuno et al. [83] and Chai et al. [84] found that in large-scale entropy, AD and MCI patients had higher entropy than NCs. Studies [85] have shown that the variation in the complexity of EEG signals associated with cognitive impairment may be inconsistent on different time scales. We normalized the results of multiscale entropy and obtained the data presented in Figure 4. In the temporal, occipitoparietal, and right frontal regions, differences were statistically significant between groups. The entropy values on a 1–20 scale in each region in the AD, MCI, and NC groups are shown in Figure 4. On short scale factors, the entropy in the NCs was greater than that in the MCI and AD patients. On long scale factors, the entropy in the AD patients was greater than that in the MCI patients, and the entropy in the MCI patients was greater than that in NC subjects. A recent study also found that, on short time scales, compared to the NC group, the AD group and MCI group had lower values of entropy and showed relative preservation of coarse-grained entropy and selective loss in fine-grained entropy. This is consistent with studies that have found lower fine-grained entropy in AD patients than in healthy older adults [86]. Perhaps these changes accompany the development of the disease from its early stage to its relatively late stage. In this case, it may be a very useful quantitative biomarker of risk. These multiscale temporal features appear to arise from functional interactions of neural structural limitations.

#### 3.1.3. Complexity Analysis in Frequency Entropy

Alqazzaz et al. [87] found that spectral results showed that EEG activity was slower in patients with AD and MCI. The SpecEn results showed that the frequency distribution of the power spectrum changed. These findings confirmed results from previous studies: the EEG signals of patients with AD and MCI gradually slowed down [76,88]. However, the physiological interpretation of all these changes is uncertain. A more scientific hypothesis is that significant brain cholinergic deficits are the basis of cognitive symptoms such as memory loss. The loss of neocortical cholinergic innervation in the modified cortex plays a key role in the EEG signal decreases associated with AD [89]. Similarly, because the cholinergic system regulates spontaneous cortical activity at low frequencies, this EEG signal decrease may also be due to the loss of the neurotransmitter acetylcholine, leading to a slowing of neural oscillations in AD. TsEn showed reductions in signal complexity in vascular dementia patients (AD) and MCI patients. In particular, the TsEn method has been shown to be a more promising complexity method for quantifying EEG changes [87,90]. Because of the speed of computation, it can serve as a theoretical basis for decision support tools in the expert diagnosis of AD [91]. Waser et al. [17] used the TsEn method to study differences between the EEGs of patients with AD and NCs and found significant differences in the t7 and t8 channels. There are also a large number of studies that have used multiple methods to explore complexity in AD. Al-Nuaimi et al. [78] found that for specific EEG frequency bands and channels, the HFD and LZC values of AD patients were significantly reduced compared to NCs. Coronel et al. [60] used automutual information (AMI), Shannon entropy, TsEn, MSE, and SpecEn to analyze the severity of AD, and the results showed that reduced complexity and AMI, SpecEn, and MSE values were associated with decreased Mini-Mental State Examination (MMSE) scores.

It is generally believed that AD leads to a decrease in high-frequency (alpha, beta, and gamma) power and an increase in low-frequency (delta and theta) power [92]. We averaged the values from five brain regions in each frequency band, resulting in the data presented in Figure 5. Figure 5 shows the differences in the frequency domain among the AD, MCI, and NC groups. On the one hand, the En value in the delta (δ), theta (θ) and gamma (γ) bands ($En_{AD}$ > $En_{MCI}$>$\;En_{Control}$) significantly increased. On the other hand, the *En* value in the alpha (α) band decreased ($\alpha En_{MCI}\;$> $\alpha En_{Control}$ > $\alpha En_{AD}$). Notably, $\alpha En_{MCI}$ was significantly higher than $\alpha En_{control}$. This result may be related to a compensatory mechanism in patients with MCI during memory load and cognitive performance; for NCs, compensation is not required, and for AD patients, compensation is no longer possible [93]. The value of  βEn was lower in the AD and MCI patients than in the NCs ($\beta En_{AD}\;$< $\beta En_{MCI}\;$< $\beta En_{Control}$).

It has been reported that different frequency bands reflect different brain dynamics [94]. We found that in applications for AD detection, the AD group showed lower complexity in different regions and sub-bands than the control group. This may be because high-frequency oscillations originate from short-range neural connections [95,96], while low-frequency oscillations include long-range neural connections [93,97]. Hence, the abnormal neural connectivity in patients with AD may be related to the abnormal complexity at different frequencies. Both the process of aging and the development of dementia has been associated with these low-frequency band increases [96]. This is partly due to the increasing local (rather than distributed) nature of the interactions between neuronal populations [98].

#### 3.1.4. Complexity Analysis in Other Methods

Jeong et al. [99] found that in most EEG channels, AD patients had significantly lower FD values than NCs. In the detection of dementia, previous studies used the FD of the correlation dimension and HFD and found that the value of FD was lower in AD patients in the parietal and temporal regions compared to NCs [16,100]. Amezquita-Sanchez et al. [101] used box dimension (BD), HFD, Katz’s FD (KFD), and the integrated multiple signal classification and empirical wavelet transform (MUSIC-EWT) to diagnose MCI and AD patients with an accuracy of 90.3%. Al-Nuaimi et al. [102] studied HFD in EEGs for AD diagnosis, and they found that HFD is a promising EEG biomarker that can capture changes in the areas of the brain that are initially affected by AD. McBride et al. [103] researched complexity based on the LZC method to distinguish patients with early MCI (EMCI), AD, and NC, and they found that the EEG complexity features of specific bands with regional electrical activity provided promising results in distinguishing EMCI, AD, and NC. Liu et al. [77] used LZC and multiscale LZC methods for analysis and found significant differences between groups in the alpha-band in the parietal and occipital regions. Hornero et al. [74] used LZC to analyze EEGs and MEGs in patients with AD and found that LZC provides good insight into the characteristics of EEG background activity and the changes associated with AD. Through these studies, we found that the HFD and LZC of the EEG are potentially good biomarkers of AD diagnosis, as they are significantly lower in AD patients than in NCs.

#### 3.1.5. Identification of AD

In this section, Table 2 shows the sensitivity, specificity, and accuracy in differentiating among AD, MCI, NC subjects were found with different nonlinear methods used in the EEG.

### 3.2. Complexity Analysis of MEG in AD and MCI

In this section, we review the signal complexity of the MEGs in MCI and AD patients compared with NC participants. The temporal resolution of MEG signals can reach the millisecond level, and the spatial resolution can be less than 2 mm. We found that the research could be generally divided based on the analysis of different brain regions to identify trends in these values. MEG is a noninvasive technique that allows recording of the magnetic fields generated by brain neuronal activity. MEG signals are independent of any reference point and are less affected by extracerebral tissues than EEG signals [106,107].

#### 3.2.1. Complexity Analysis in Domain Entropy

Gómez et al. [108,109] analyzed MEG complexity based on cross-approximate entropy, which revealed decreases that indicated better synchronization in AD and MCI patients than in NC subjects. Using the ApEn, SampEn, and FuzzyEn methods to analyze MEG signals at 148 locations, it was found that the entropy in AD patients was lower than that in controls, suggesting that this neurological disorder may be accompanied by a regular increase in MEG activity. Hornero et al. [86] found that ApEn, SampEn, and MSE values in MEG data were lower in AD patients than in NCs. Juan P et al. [110] found that all PeEn values in the MCI group were larger than those in the normal group. Azami et al. [111] used the FuzzyEn, SampEn, and PeEn methods, and a 148-megabyte channel was analyzed to quantify the complexity of the signal. The FuzzyEn and SampEn values in AD patients were lower than those in the controls. AD patients showed significantly lower values than MCI subjects and NCs in almost all comparisons. Most studies have yielded information about the location of similar brain regions. Gómez et al. [112] reported MSE profiles that represented the SampEn values of each coarse-grained time series relative to the scale factor. Azami et al. [113] found that the values of multiscale dispersion entropy (MDE), multiscale permutation entropy (MPE), and MSE in AD patients were lower than those in NCs at short scale factors, while at long scale factors, the MDE and MSE values from AD subject signals had higher values [112]. In contrast, the MPE values at long scale factors were very similar for AD patients and NCs.

We found that most of the studies were divided based on the analysis of different brain regions and were analyzed on different scales. At low scale factors, the entropy value in AD patients was lower than that in NCs. For high scale factors, the values in AD patients were higher than those in controls. Figure 6 shows data for each region, and we report the average entropy values computed across the entire 1–20 scale factor range. In terms of the MEG signal, AD patients were reported to be more regular, less complex and more predictable than the controls, and these results were consistent with the EEG results [114].

#### 3.2.2. Complexity Analysis in Frequency Entropy

Nonlinear analysis of frequency has also been reported in MEG-based studies by SpecEn and ratios. Poza et al. [115] studied the ratio of SpecEn (RSP). In the delta and theta bands, the RSP in AD patients was significantly higher than that in controls. However, in the beta and gamma bands, the RSP value was significantly lower in AD patients than in NCs. Regarding the spectral entropies, the results showed a statistically significant decrease in the value in the MEG signal in AD patients compared to NCs. Poza et al. [116] found that the spectral crest factor and both spectral turbulence and wavelet turbulence in AD patients were higher than those in NCs, which indicated that in AD patients, the oscillating signal was more regular. Bruner et al. [117] found that the SpecEn and TsEn values in patients with MCI were significantly lower than those in controls in the right lateral region, indicating a significant decrease in the irregularity of MEG signals in patients with MCI. All studies have shown that AD patients had slower brain activity than controls, which was reflected in a higher power in the lower frequency bands and lower power in the higher frequency bands.

#### 3.2.3. Complexity Analysis in Other Methods

Gómez et al. [118] researched MEG background activity from AD and NC subjects using HFD and found that the value of HFD was less complex in AD patients, indicating an abnormal type of motility in AD. Shumbayawonda et al. [119] used LZC to research MEG signals in three groups: NCs, patients with subjective cognitive decline (SCD), and patients with MCI, and analyses were performed in theta, alpha, beta, and gamma bands. It was found that the LZC value in MCI patients was significantly lower than that in the control group and in SCD subjects, and the lower complexity was associated with smaller hippocampal volume. Another study, combining age with LZC, found that AD patients and controls showed a tendency of decreased LZC with age [120]. We found that both non-entropy and entropy methods for assessing complexity achieve the same results, but entropy methods were more widely used.

#### 3.2.4. Identification of AD

In this section, Table 3 shows the sensitivity, specificity, and accuracy in differentiating among AD, MCI, NC subjects were found with different nonlinear methods used in the MEG.

### 3.3. Complexity Analysis of fMRI and fNIRS Signals in AD and MCI

The fMRI uses magnetic array imaging [127,128], while fNIRS uses hemoglobin in blood vessels to scatter near-infrared light [129,130]. In this section, we review signal complexity from fMRI and fNIRS in MCI and AD patients compared with NCs. A few studies have reported that biomarkers from fMRI and fNIRS signals, such as LZC, entropy, and other complexity characteristics, differ between MCI, AD, and NC subjects.

The fMRI signals have been used to detect functional abnormalities associated with neuropsychiatric and neurological disorders. Maxim et al. [131] applied the HE method to fMRI signals, and they found that the values of signals in the white matter were lower than those in the gray matter. Liu et al. reported that the complexity in certain brain regions (e.g., anterior cingulate gyrus and left cuneus) was reduced in a study of resting-state fMRI (rs-fMRI) signal complexity in familial AD patients [132]. Wang et al. [15] found significantly decreased PeEn values in the AD patient group compared with the MCI group and the normal group. Compared with the NC group, the complexity in the left wedge in the MCI group was also reduced. The complexity differences among the groups were mainly observed in the temporal, occipital, and frontal lobes. We found that AD patients had reduced mean whole-brain complexity in the gray matter and white matter compared to EMCI and NC subjects. At the regional level, five clusters showed significant differences in En, as illustrated in Figure 7. Niu et al. [133] extracted the average MSEs of the whole brain, gray matter, white matter, and cerebrospinal fluid using corresponding masks on all time scales. Only the gray matter showed a trend toward an entropy difference between the groups at scale factor six. Significant differences were found between the groups at scale factors two, four, five, and six, as shown in Figure 8. A significant difference was found in the right thalamus at scale factor two. A significant difference was found in the left superior frontal gyrus at scale factor four. Two significant differences were found in the right lingual gyrus and right insula at scale factor five. Five significant differences were found in the right superior temporal gyrus, left middle temporal gyrus, right olfactory cortex, left inferior occipital gyrus, and right supramarginal gyrus (SMG.R) at scale factor six. Grieder et al. [134] found that the AD group showed a lower global default mode network (DMN)-MSE than the NC group. A scientific explanation has been found for the reduced complexity of fMRI and fNIRS signals in AD. High regional functional homogeneity leads to lower complexity, so more differentially affected brain regions are found at high scale factors. Nerve cells are associated with complex dynamic processing in brain neural networks, and neuronal cell death leads to the loss of connectivity to local neural networks. It may be the death of neurons and the lack of neurotransmitters that lead to reduced irregularity in AD patients.

## 4. Discussion

Complexity methods applied to brain imaging data such as EEG, MEG and fMRI provide useful information for the diagnosis of AD using abnormal brain activity signals. This review combines previous findings with a larger overview and a further characterization of multiple modes for a better understanding of the functional lesion in AD. For the complexity of the single-channel time series, the development process of AD is clear and independent of the method used. The decline in AD may be due to plaques and cell death leading to loss of connectivity between cortical neurons, which may lead to more regular brain signals, thus destroying effective communication throughout the brain [135,136]. Furthermore, for each part of the brain, the trend is not consistent [137]. This may be related to the compensatory mechanisms that exist in the brain: when the synaptic structure slows down less, new synapses can be established to fill the gap, to change the connection pathway and establish connections with other regions or to increase the degree of added work, thus compensating for the altered brain function compensate, which indicates that the complexity of the AD brain changes [138,139]. The pattern of changes in this complexity is a good reflection of the pathological progression of AD and shows that complexity can be used as a biomarker to measure AD.

Complexity methods are suitable for the study of nonlinear brain changes and are sensitive to neurological changes associated with AD patients compared with normal subjects. The entropy method accounts for a large proportion of the complexity methods used. Better performance was exhibited at high scales, and when more brain regions were included in the analysis, the trends were more obvious. The exploration of test–retest reliability and improvements in entropy algorithms will provide great guidance for future applications. As the brain is a complex system in time and space, we can also study network entropy and spatio-temporal entropy in the future. While fMRI has a spatial resolution on the order of millimeters, only a small number of studies have applied complexity to fMRI data to date. Although the time resolution is not very high, it also reacts well and has been used to identify downward trends in different brain regions. It is also important to note that the potential utilization of the high spatial resolution in fMRI and fNIRS data can provide more in-depth information for AD brain dysfunction.

Complexity analysis of different types of brain imaging data in AD patients has yielded consistent results. The results showed consistent changes in that the signals in the brains of AD patients are slower, more regular, less complex, and less well organized than those of NCs. The reduction in the irregularity and complexity of brain signals in AD is the main finding obtained, and the occipital, frontal, parietal, and temporal areas are the most affected regions. We found that complexity can capture changes in areas of the brain that are initially affected by AD and accurately respond to its pathological mechanism. Complexity is a promising biomarker in reflecting the pathological mechanism of AD, and entropy is the more widely used of the numerous complexity indicators described in this review. For which entropy index is the best, more research is needed in the future to prove it. In general, different modalities for the same groups with large amounts of data were analyzed by choosing methods with high reliability and accuracy, the results of which will aid in truly understanding the functional lesion in AD. The results of the articles in this review can advance research on quantifying the complexity indexes of different subjects until clinical application is realized.

## Figures and Tables

**Figure 1 entropy-22-00239-f001:**
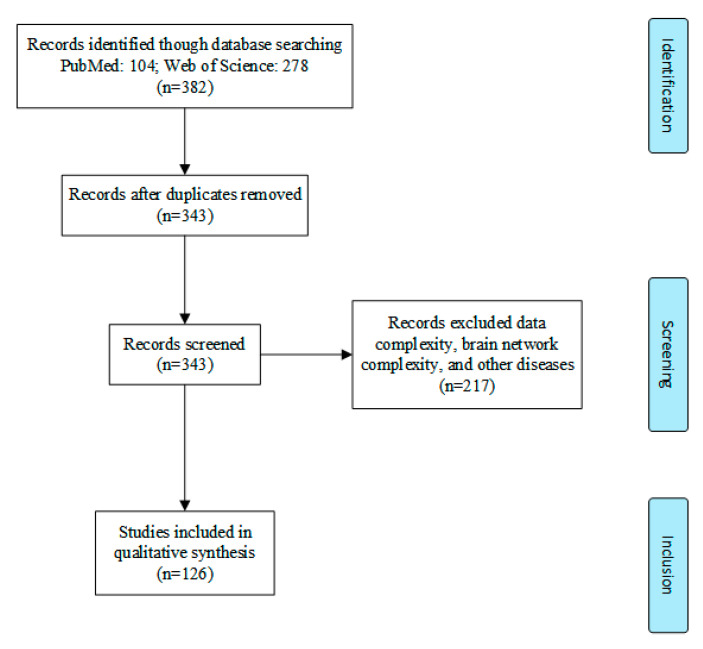
Selection diagram, including three stages: identification, screening, and inclusion. This process led from 382 initial studies to 126 final studies.

**Figure 2 entropy-22-00239-f002:**
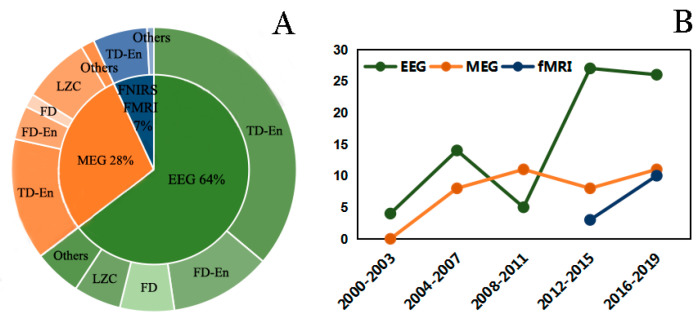
(**A**) Three modes of data categorization reviewed in the study. The inner circle shows the different brain imaging modalities, while the outer circle shows specific complexity analysis methods. (**B**) Trends in the number of included studies using the different brain imaging techniques versus date.

**Figure 3 entropy-22-00239-f003:**
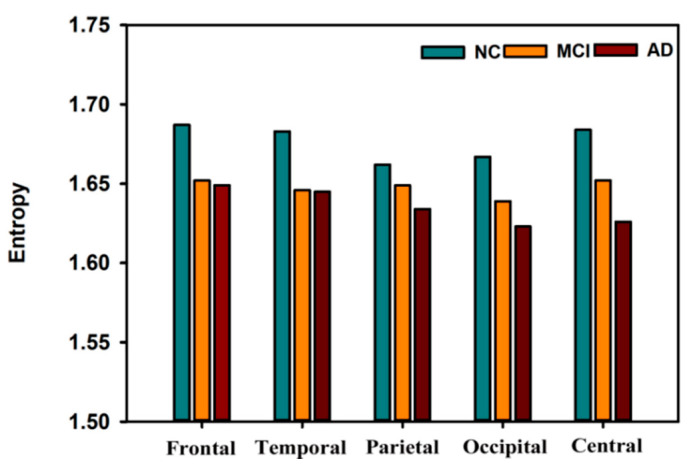
Comparative values of entropy from five regions across the brain in Alzheimer’s disease (AD), mild cognitive impairment (MCI), and control subjects [18,81].

**Figure 4 entropy-22-00239-f004:**
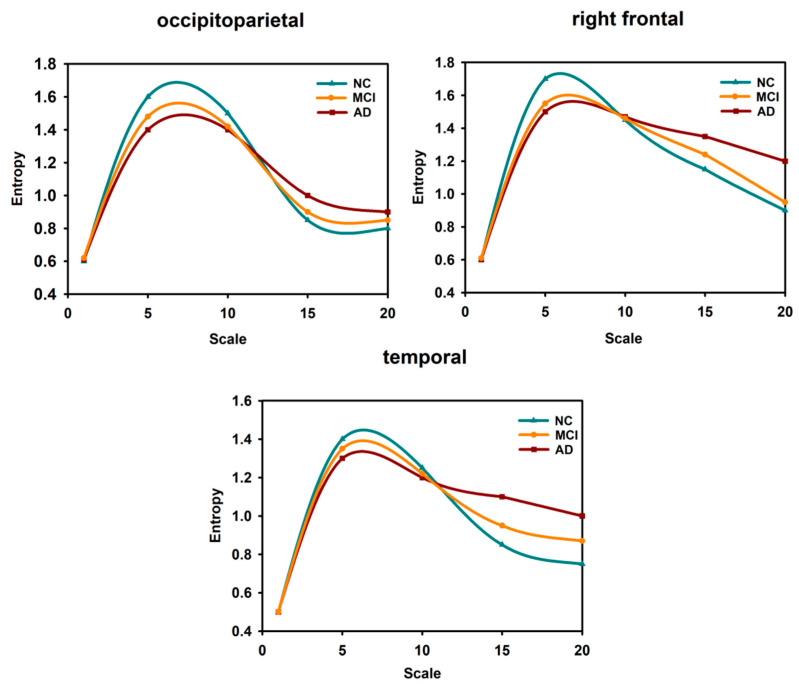
Entropy at different scales in different regions of the brain [82,84].

**Figure 5 entropy-22-00239-f005:**
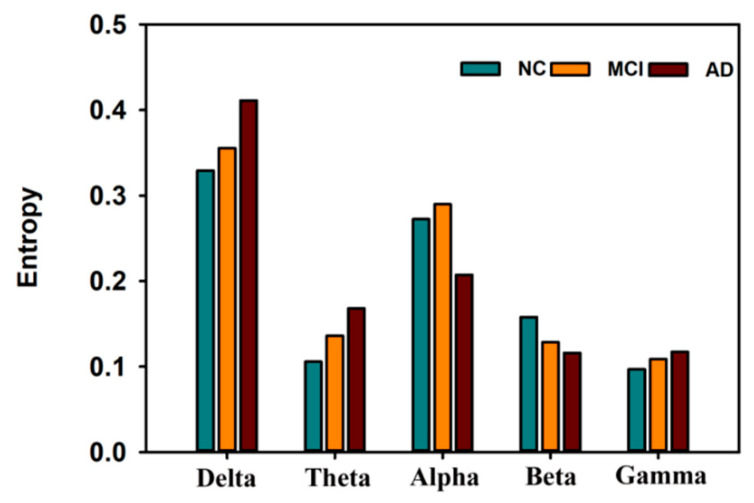
Entropy in different frequency bands across the brain in AD, MCI, and control subjects [81].

**Figure 6 entropy-22-00239-f006:**
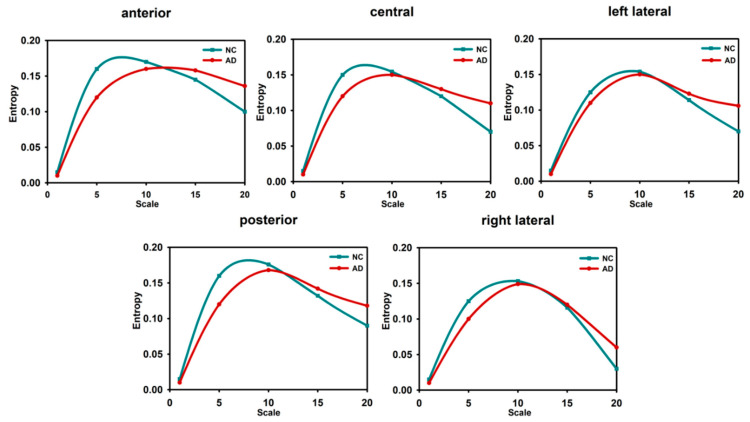
Entropy from different brain regions [48,113].

**Figure 7 entropy-22-00239-f007:**
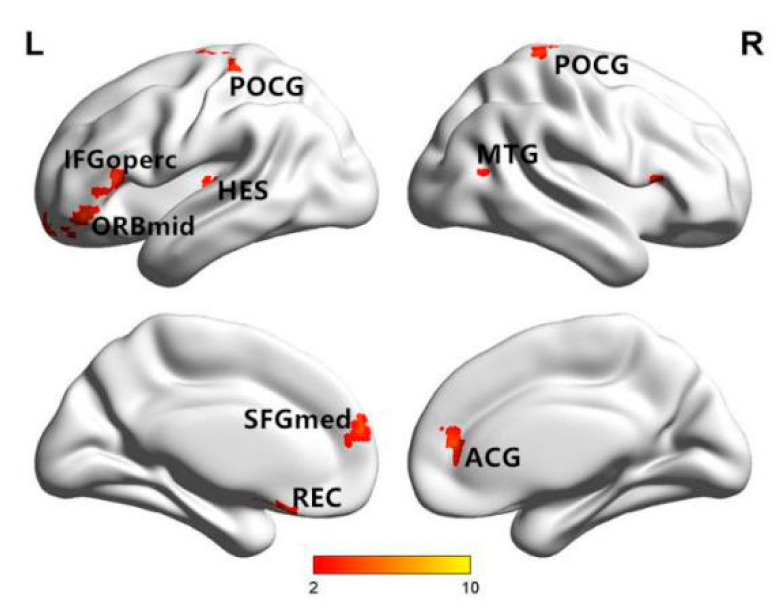
Brain regions with significant differences between groups [15].

**Figure 8 entropy-22-00239-f008:**
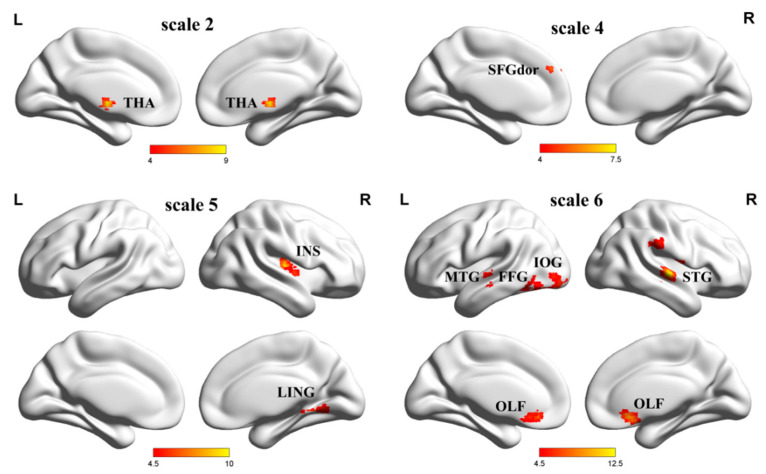
Brain regions with significant differences between groups on scale factors two, four, five, and six [133].

**Table 1 entropy-22-00239-t001:** Summary of widely used complexity analysis methods.

	Complexity Indices	Abbreviations	Year	Description
Time domain entropy	Approximate entropy	ApEn	Pincus (1991) [51]	Needs only a small dataset and is effective for discriminating the signal from random signals. A higher value indicates more irregularity.
	Sample entropy	SampEn	Richman (2000) [52]	The exact value of the negative average natural logarithm of the conditional probability. A higher value indicates less predictable signals.
	Permutation entropy	PeEn	Bandt (2002) [27]	Only considers the grades of the samples but not their metrics. A higher value indicates a more irregular signal.
	Multiscale entropy	MEn	Costa (2005) [53]	Can be observed at multiple different scales of signal change.
	Fuzzy entropy	FuzzyEn	Chen (2007) [54]	Provides a mechanism for measuring the degree to which a pattern belongs to a given class.
Frequency domain entropy	Renyi entropy	ReEn	Renyi (1977) [55]	Forms the basis of the concept of generalized dimensionality. If the Renyi entropy is high, the signal has high complexity.
	Spectral entropy	SpecEn	Powell (1979) [56]	Predictability according to an analysis of the spectral content of a signal. A high value indicates a more irregular and less predictable signal.
	Tsallis entropy	TsEn	Tsallis (1998) [57]	Explores the properties of a probability distribution from a new mathematical framework.
Others	Hurst exponent	HE	Hurst (1951) [58]	Used to measure the long-term memory and fractal dimension of a time series.
	Lempel-Ziv complexity	LZC	Lempel (1976) [59]	Reconstructs the original time series into a binary sequence. A high value indicates a high variation in the binary signal.
	Correlation dimension	D2	Grassberger (1983) [60]	The number of independent variables needed to describe the time series dynamics after the time series is transferred to chaos theory-based state space.
	Fractal dimension	FD	Higuchi (1988) [61]	It complements the chaos theory of the dynamic system, showing the similarity with the whole.

**Table 2 entropy-22-00239-t002:** Sensitivity, specificity, and accuracy in differentiating among AD, MCI, and normal control (NC) subjects were found with different nonlinear methods used in the electroencephalogram (EEG) database (NR represents that the paper does not give this value accurately).

Research	Method	Class	Sensitivity	Specificity	Accuracy	AUC
Sharma et al. (2019) [88]	SpecEn + FD	NC vs. MCI	86%	81%	84.1%	NR
		MCI vs. AD	83%	63%	73.4%	NR
		NC vs. AD	82%	82%	82%	NR
Chai et al. (2019) [84]	MSE	NC vs. MCI	NR	NR	NR	73%
		NC vs. AD	NR	NR	NR	81%
Fan et al. (2018) [104]	MSE	NC vs. AD	88.71%	69.09%	79.49%	83%
Houmani et al. (2018) [105]	EpEn (epoch-based entropy)	SCI vs. AD	87.8%	100%	91.6%	NR
Simons et al. (2018) [75]	ApEn	NC vs. AD	90.91%	63.64%	77.27%	NR
	SampEn		90.91%	63.64%	77.27%	NR
Al-Nuaimi (2018) [78]	ApEn	NC vs. AD	72.73%	81.82%	77.27%	85.95%
	SampEn		81.82%	72.73%	77.27%	85.95%
	LZC		81.82%	81.82%	81.82%	89.26%
	FuzzyEn		81.82%	90.91%	86.36%	86.78%
	MSE		90.91%	90.91%	90.91%	93.39%
	AMI		100%	81.82%	90.91%	93.39%
	HFD		66.67%	100%	80%	NR
Al-Qazzaz (2016) [87]	TsEn	NC vs. AD	85.71%	84.62%	85%	NR
	LZC		100%	92.31%	95%	NR
Liu et al. (2016) [77]	LZC	NC vs. AD	80.0%	78.1%	78.5%	89.21%
	MS_LZC (multiscale_LZC)		86.8%	84.3%	85.7%	91.12%

**Table 3 entropy-22-00239-t003:** Sensitivity, specificity, and accuracy in differentiating among AD, MCI, and NC subjects were found with different nonlinear methods used in the magnetoencephalogram (MEG) database (NR represents that the paper does not give this value accurately).

Research	Method	Class	Sensitivity	Specificity	Accuracy	AUC
Azami et al. (2016) [121]	MFE (multiscale fuzzy entropy)	NC vs. AD	NR	NR	78.22%	NR
Juan P. et al. (2016) [110]	PeEn	MCI vs. AD	NR	NR	98.4%	NR
Escuderoa et al. (2015) [122]	MSE	NC vs. AD	94.4%	46.2%	NR	67%
Gómez et al. (2014) [109]	SampEn	NC vs. AD	80.00%	61.90%	70.73%	NR
	LZC		80.00%	76.19%	78.05%	NR
Bruña et al. (2012) [117]	ShEn	NC vs. AD	NR	NR	69.4%	79.0%
		NC vs. MCI	NR	NR	65.9%	64.1%
		MCI vs. AD	NR	NR	64.8%	69.1%
	TsEn	NC vs. AD	NR	NR	75.8%	85.6%
		NC vs. MCI	NR	NR	61.4%	60.7%
		MCI vs. AD	NR	NR	66.7%	75.6%
	ReEn	NC vs. AD	NR	NR	83.9%	89.0%
		NC vs. MCI	NR	NR	63.6%	65.2%
		MCI vs. AD	NR	NR	72.2%	78.5%
Poza et al. (2012) [123]	SampEn	NC vs. AD	88.9%	57.7%	75.8%	80.6%
Gómez et al. (2010) [124]	SampEn		77.78%	50.00%	66.13%	71.26%
	ApEn		75.00%	53.85%	66.13%	73.82%
	HFD		72.22%	73.08%	72.58%	79.11%
	LZC		80.56%	61.54%	72.58%	78.63%
	ShEn		91.67%	57.69%	77.42%	79.27%
Hornero et al. [74]	ApEn	NC vs. AD	75.0%	66.7%	70.7%	NR
	AMI		75.0%	90.5%	82.9%	NR
	LZC		85.0%	85.7%	85.4%	NR
Gómez et al. (2007) [112]	SampEn	NC vs. AD	80%	76.2%	NR	84%
	MSE		75%	100%	NR	87.8%
Poza et al. (2008) [125]	ShEn	NC vs. AD	85.0%	81.0%	82.9%	NR
	ReEn		90.0%	85.7%	87.8%	NR
Hornero et al. (2008) [126]	ApEn	NC vs. AD	50.0%	52.4%	51.2%	NR
	LZC		65.0%	76.2%	70.7%	NR
	SpecEn		70.0%	76.2%	73.2%	NR

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
