# Peer review of "Complexity Analysis of EEG, MEG, and fMRI in Mild Cognitive Impairment and Alzheimer’s Disease: A Review"

_entropy, 2020, doi:10.3390/e22020239_

Round 1
Reviewer 1 Report
The article presents a review of papers on complexity-based features extracted from different recordings of brain activity of the subjects with the Alzheimer's disease.
Comments:
Abstract should be more specific. Indicate the time period of reviewed articles. Summarize main findings. Structural organization of the paper is poor. Move description of search query from Introduction to Methods section. Last sentence of the introduction section discusses the results, this would better fit the Discussion or Conclusion section. Following a good practice of IMRAD template is suggested. In the introduction section also discuss the related review articles, which also analyse and discuss the works on brain analysis, such as A survey on computer-assisted parkinson's disease diagnosis. Artificial Intelligence in Medicine, 95, 48-63. doi:10.1016/j.artmed.2018.08.007 Add the Methods section to describe the methodological principles of the review process. Which guidelines of systematic review you follow. See for example, PRISMA (Preferred Reporting Items for Systematic Reviews and Meta-Analyses). I did not find any research questions or hypotheses formulated. What are you trying to find by your review. As a result, the presentation is rather chaotic and unfocused, while the findings are vague. Figure 2: explain all abbreviations used in the caption. Figure 5 caption: En should be entropy? Explain all abbreviations used in the figure. Figures 3-6 present the results of some stand-alone studies on a specific selection of subjects and stimuli applied. The review paper rather should compare and summarize the results of multiple studies. The search query you use omits many important works in the area. Authors may not necessarily use the “Complexity analysis” OR “Nonlinear dynamical analysis” keywords to describe their work, but still use, for example, Hurst complexity, fractal dimension, fuzzy entropy or other complexity-related features. As a result you miss many important works in the are which should have been included in your analysis, for example: A novel methodology for automated differential diagnosis of mild cognitive impairment and the alzheimer's disease using EEG signals. Journal of Neuroscience Methods, 322, 88-95. doi:10.1016/j.jneumeth.2019.04.013 Automated detection of Alzheimer’s disease using brain MRI images– A study with various feature extraction techniques. Journal of Medical Systems, 43(9) doi:10.1007/s10916-019-1428-9 Early alzheimer's disease diagnosis based on EEG spectral images using deep learning. Neural Networks, 114, 119-135. doi:10.1016/j.neunet.2019.02.005 Adaptive independent subspace analysis of brain magnetic resonance imaging data. IEEE Access, 7, 12252-12261. doi:10.1109/ACCESS.2019.2893496 Discuss the reliability of your results and threats-to-validity. Discuss, for example, the influence of Inappropriate or incomplete search terms in automatic search, selection of article databases, selection of inclusion & exclusion criteria, restricted time span, etc. The conclusion should present and summarize your answers to the research questions, which however were never formulated. Be more specific in the presentation of your main findings. Avoid trivial propositions such as “Complexity methods are suitable for the study of nonlinear brain changes”.Author Response
Thank you very much for your serious and rigorous comments.The following modifications are made in this manuscript:
Abstract section:The time period of reviewed articles has been added, and consistency conclusions for the literature have been drawn. The aim of this review is to promote an understanding of AD and to investigate whether signal complexity can be used as a biomarker to accurately respond to the AD lesion process. Introduction section:The search strings from the Introduction section have been moved to the Methods section, and the sentence elaborating on the results have been moved to the Discussion section. The introduction has been modified to illustrate the necessity of summarizing the different modalities and now explains the complexity abnormalities in the AD brain. Following the IMRAD template, the article consists of the Introduction, Methods, Results, and Discussion sections. Abbreviations were checked and corrected, and Table 1 also adds a list of the abbreviations. We followed the PRISMA (Preferred Reporting Items for Systematic Reviews and Meta-Analyses) guidelines. Our research focuses on the complexity of single channel time series. The keywords have been updated: (“Complexity analysis” OR “Nonlinear dynamical analysis” OR “Lempel-Ziv complexity” OR “fractal dimension” OR “hurst exponent” OR “entropy” OR “correlation dimension”) AND (“Alzheimer’s disease” OR “Mild Cognitive Impairment” OR “Subjective Cognitive Impairment”). The number of articles increased from 377 to 382, and the number of selected documents was 126. Most of the 16 additional articles involved the application of the Lempel-Ziv complexity (LZC) and fractal dimension (FD) methods, which have been added accordingly in the Results section. Figure 1 and Figure 2 have been updated. Figures 3-6 are summaries of the results described by a large number of references. They were generated by the authors of this review. Research the main problem solved: Promote an understanding of AD; 2. Include future research directions; 3. Whether complexity can be used as biomarkers to accurately respond to the AD lesion process. The Abstract, the Introduction and Discussion section have been modified accordingly.
Reviewer 2 Report
The time period of reviewed articles should be clarified also in the abstract.
The authors used the following string of keywords: “Complexity analysis” OR “Nonlinear dynamical analysis”. I wonder if the number of selected articles will change if also other keywords are used. As an example: "network analysis", fuzzy analysis" or entropy. If a new search (and consequent analysis) will not be performed, the authors should at least discuss this issue in their revised discussion
According to PRISMA (Preferred Reporting Items for Systematic Reviews and Meta-Analyses). research questions or hypotheses should be formulated. These lack in the current version of the Ms.
Please, avoid to use nondefined abbreviations
Author Response
Thank you very much for your serious and rigorous comments.The following modifications are made in this manuscript:
Abstract section: The time period of the reviewed articles has been added. Our research focuses on the complexity of single channel time series.The keywords have been updated: (“Complexity analysis” OR “Nonlinear dynamical analysis” OR “Lempel-Ziv complexity” OR “fractal dimension” OR “hurst exponent” OR “entropy” OR “correlation dimension”) AND (“Alzheimer’s disease” OR “Mild Cognitive Impairment” OR “Subjective Cognitive Impairment”). The number of articles increased from 377 to 382, and the number of selected documents was 126. Most of the 16 additional articles are for the application of the Lempel-Ziv complexity (LZC) and fractal dimension (FD) methods, which have been added accordingly in the Results section. We followed the PRISMA (Preferred Reporting Items for Systematic Reviews and Meta-Analyses) guidelines; this has been added to the text. Research the main problem solved: 1. Promote an understanding of AD; 2. Include future research directions; 3. Whether complexity can be used as biomarkers to accurately respond to the AD lesion process. The Abstract, Introduction and Discussion section have been supplemented accordingly. Abbreviations were checked and corrected, and Table 1 also adds a list of the abbreviations.Round 2
Reviewer 1 Report
Abstract: what is "AD lesion process"? It is mentioned only in the abstract.
L. 64-65: "We aim to outline a complexity index that can accurately represent the pathological mechanism of AD and to study which of the many complexity indicators is best." -> I do not think that the article fully answers that research question. So which of many entropy metrics listed in Table 2 or Table 3 is the best?
Table 1, column "Year": add reference numbers to the papers cited.
Table 2: the numbers in the last row miss the percentages.
Table 3: explain the abbreviation "NR".The last row mixes absolute values and percentage values.
Author Response
Thank you very much for your valuable comments. We have made the following revised to your comments:
Abstract: thank you very much for you good comment. We change "AD lesion process" to "functional lesion in AD”, and in the text unified statement. The current complexity studies were mainly focused on the abnormal in the brain area that showed functional lesion in AD. To complete outline the AD lesion process by the way of complexity, more research to support, more in-depth and extensive research to discover the lesion process with the discovery of the brain area with functional lesion. So, in current review, we thought the functional lesion in AD is more suite for the state of complexity studies of AD.
L.64-65: Change to " We aim to review the complexity indexes those can accurately represent the lesion brain area of AD and to outline the better complexity indicators.” We think the entropy index performs better in many complexity indexes. The entropy method accounts for a large proportion of complexity methods used. Better performance was exhibited at high scales, and when more brain regions were included in the analysis, the trends were more obvious. But for which of the many entropy metrics is the best one, more research conclusions are indeed needed to support in the future studies.
Table 1, column "Year" with references added.
Table 2: Percentage added to the table.Add paper year in the first column.And the tables are sorted by year.
Table 3: An explanation of "NR" is added to the header and the last line is modified to a percentage value.Add paper year in the first column.And the tables are sorted by year.

Reviewer 2 Report
The authors have responded to the points I raised.
In my opinion, it is now acceptable for publication
Author Response
Thank you for your agreement.